# Remdesivir Influence on SARS-CoV-2 RNA Viral Load Kinetics in Nasopharyngeal Swab Specimens of COVID-19 Hospitalized Patients: A Real-Life Experience

**DOI:** 10.3390/microorganisms11020312

**Published:** 2023-01-25

**Authors:** Laura Campogiani, Marco Iannetta, Andrea Di Lorenzo, Marta Zordan, Pier Giorgio Pace, Luigi Coppola, Mirko Compagno, Vincenzo Malagnino, Elisabetta Teti, Massimo Andreoni, Loredana Sarmati

**Affiliations:** 1Department of System Medicine, Tor Vergata University, 00133 Rome, Italy; 2Infectious Disease Clinic, Policlinico Tor Vergata, 00133 Rome, Italy

**Keywords:** SARS-CoV-2, coronavirus, remdesivir, antiviral, viral load, cycle threshold, nasopharyngeal swab

## Abstract

There are still conflicting data on the virological effects of the SARS-CoV-2 direct antivirals used in clinical practice, in spite of the documented clinical efficacy. The aim of this monocentric retrospective study was to compare virologic and laboratory data of patients admitted due to SARS-CoV-2 infection from March to December 2020 treated with either remdesivir (R), a protease inhibitor (lopinavir or darunavir/ritonavir (PI)) or no direct antiviral drugs (NT). Viral load variation was indirectly assessed through PCR cycle threshold (Ct) values on the nasopharyngeal swab, analyzing the results from swabs obtained at ward admission and 7 (±2) days later. Overall, 253 patients were included: patients in the R group were significantly older, more frequently males with a significantly higher percentage of severe COVID-19, requiring more often intensive care admission, compared to the other groups. Ct variation over time did not differ amongst the three treatment groups and did not seem to be influenced by corticosteroid use, even after normalization of the treatment groups for disease severity. Non-survivors had lower Ct on admission and showed a significantly slower viral clearance compared to survivors. CD4 T-lymphocytes absolute count assessed at ward admission correlated with a reduced Ct variation over time. In conclusion, viral clearance appears to be slower in COVID-19 non-survivors, while it seems not to be influenced by the antiviral treatment received.

## 1. Introduction

In December 2019, a new type of *Betacoronavirus* emerged in Wuhan, China. It was defined as Severe Acute Respiratory Syndrome Coronavirus 2 (SARS-CoV-2), and the disease it causes was named coronavirus disease 2019 (COVID-19) [1,2]. Symptoms related to SARS-CoV-2 infection may vary from asymptomatic disease to severe respiratory illness, requiring hospitalization and mechanical ventilation [3].

Viral nucleic acid detection by real-time reverse transcription polymerase chain reaction (RT-PCR) on a nasopharyngeal (NPh) swab is considered the gold standard for the etiological diagnosis of COVID-19, and the number of PCR cycles required for the fluorescent signal to cross the background level (threshold) is defined as cycle threshold (Ct) [4]. Viral load in respiratory specimens has been extensively investigated, together with host immunological factors. Ct has often been used as a proxy of viral load, with an inverse correlation [5]. It has been shown that SARS-CoV-2 nasopharyngeal Ct values correlate with viremia at admission, and an increased risk of 60-day mortality has been described in patients with low nasopharyngeal Ct values at infection onset [6]. Kurzeder et al. showed that on hospital admission, COVID-19 patients with nasopharyngeal Ct values ≤26 had an increased risk of in-hospital death [7]. SARS-CoV-2 nasopharyngeal Ct values, together with age, comorbidities, and severity scores, can help identify COVID-19 patients with a severe prognosis, also in the intensive care setting [8]. Furthermore, patients with more severe disease have a slower viral clearance [9].

As for therapies, throughout the pandemic, different therapeutic regimens have been evaluated and used, targeting both the virus itself and the host immune system. Amongst direct antivirals, protease inhibitors lopinavir and darunavir combined with ritonavir have been used in the early pandemic phases based on some evidence of in vitro and in silico activity of these compounds against SARS-CoV-2 replication [10] but then rapidly abandoned. Remdesivir, which is a prodrug converted to an adenosine nucleoside triphosphate analog, acts as an irreversible chain terminator, blocking transcription by the viral RNA polymerase of several viruses, including coronaviruses, Ebola, and hepatitis C virus. This antiviral agent demonstrated potent inhibition of SARS-CoV-2 replication in vitro [11,12]. Therefore, it received emergency approval for COVID-19 treatment from the Food and Drug Administration in May 2020. The Adaptive COVID-19 Treatment Trial-1 (ACTT-1) demonstrated that remdesivir was superior to placebo in shortening the time to recovery in adults hospitalized for COVID-19 with evidence of lower respiratory tract infection [13]. Based on this finding, remdesivir has been widely used to treat COVID-19. Up-to-date remdesivir is recommended as early treatment (short 3-day treatment course) in fragile subjects at high risk for hospitalization, together with the oral antivirals nilmatrevir/ritonavir and molnupiravir. Remdesivir is also the only intravenous direct antiviral recommended to treat both non-severe and severe COVID-19 hospitalized patients (5 to 10 days treatment) [14,15,16,17].

In vivo evidence suggests that remdesivir is associated with reduced time to symptom resolution [18], although the impact on other clinical outcomes such as mortality, initiation of ventilation, and duration of hospital stay remains uncertain [18,19,20,21,22,23]. The latest international COVID-19 treatment guidelines report a reduction in mortality and need for hospitalization in patients treated with remdesivir, with a focus on early administration in high-risk outpatients [15,16,17]. Studies focusing on the effect of remdesivir on viral outcomes, including viral load and clearance from respiratory specimens, still show contrasting results [9,24,25]. A retrospective case-control study on a small number (45) of hospitalized COVID-19 patients and a retrospective cohort study on 86 severe COVID-19 patients showed an effect of remdesivir in significantly reducing SARS-CoV-2 viral load on nasopharyngeal swabs [26,27]. Conversely, a retrospective study on 142 hospitalized COVID-19 patients did not show any effect of remdesivir on SARS-CoV-2 nasopharyngeal viral load compared with non-treated patients [28].

The aim of this study was to assess the effect of remdesivir on viral decay on NPh swabs, evaluated through Ct variation over time in hospitalized patients treated with this antiviral, compared with patients who received protease inhibitors or no antiviral treatment.

## 2. Materials and Methods

The present study is a single-center, retrospective, observational study performed at the Policlinico Tor Vergata University Hospital of Rome, Italy, involving patients hospitalized for SARS-CoV-2 infection. Adult (≥18 years) patients admitted to the Infectious Disease (ID) Clinic of Policlinico Tor Vergata University Hospital from 5 March to 30 July 2020, and from 2 September to 31 December 2020, with a positive reverse transcription RT-PCR for SARS-CoV-2 on an NPh swab, were included. The study was approved by the local Ethics Committee (experimentation register number 154/21) and conducted in accordance with the principles of the Declaration of Helsinki. Given the retrospective nature of the study, patients’ written informed consent was not required.

The GeneFinderTM COVID-19 Plus RealAmp Kit, ELITech AllplexTM 2019-nCoV Assay (Seegene), was used for Real Time-PCR. It is based on the identification of three viral genetic targets: Envelope (E), Nucleocapsid (N) and RNA-dependent RNA-Polymerase (RdRP) genes. The cycle threshold (Ct) obtained from the RT-PCR is used as a proxy of the actual viral load and is inversely proportional to SARS-CoV-2 viral load. Subjects with RT-PCR cycle threshold (Ct) of each gene on the first NPh swab (T0) performed at ID ward admission were included. The study population was then restricted to subjects with NPh swabs repeated 7 (±2) days after the first NPh swab (T7), with Ct reported.

The included patients were classified according to the treatment received into three groups: PI group if they received lopinavir/ritonavir and/or darunavir with or without ritonavir; R group if they received remdesivir; NT if no antiviral treatment was administered. Patients that received interleukin (IL)-6 inhibitors were excluded.

Patients were further stratified into two groups: non-survivors if death occurred during hospitalization (in-hospital mortality), and survivors, who were either (1) discharged home, (2) moved to a residential structure for COVID-19, being medically stable because of public health issues, (3) moved to a different medical ward, in good condition (4) moved to another hospital and were still alive 30 days after the first hospitalization. Finally, patients were divided according to the oxygen supply/ventilation support required during the hospitalization, considering the highest support needed. Five groups were identified: ambient air (AA), Venturi oxygen mask (VMK), non-rebreather oxygen mask with concentrator (NRM), non-invasive ventilation (NIV) and invasive mechanical ventilation through orotracheal intubation (OTI). Subsequently, these categories were additionally grouped into non-severe (AA and VMK) and severe (NRM, NIV and OTI).

Demographics, clinical and laboratory data and peripheral blood lymphocyte subset counts were retrospectively collected in an ad hoc created database and analyzed. Viral decay was evaluated through Ct variation analysis, comparing the results of RT-PCR assays for SARS-CoV-2 RNA detection at T0 and T7 NPh swabs and calculating Ct variation for each gene (delta Ct = Ct T7–Ct T0). To allow the mathematical operation, genes undetectable at RT-PCR were arbitrarily assigned a Ct value of 45.

### Statistical Analysis

Differences between groups were assessed using the non-parametric Mann–Whitney U test (two groups, continuous variable), Kruskal–Wallis test (more than two groups, continuous variable), or the Chi2 test (categorical variables). Linear correlation was assessed using Spearman’s correlation test. The cutoff values of Ct were determined by the receiver operator characteristics (ROC) analysis and the Youden criterion.

Results were considered statistically significant if the *p*-value was lower than 0.05. Statistical analyses were performed using the software JASP (Version 0.11.1, JASP Team, 2019, Amsterdam, The Netherlands) and Prism 8 for macOS (version 8.2.1, GraphPad Software, San Diego, CA, USA).

## 3. Results

Five hundred and thirteen patients hospitalized for SARS-CoV-2 infection in the Infectious Disease Clinic of Policlinico Tor Vergata University Hospital of Rome (Italy) from 5 March to 30 July 2020, and 2 September to 31 December 2020, were enrolled in the study. Four hundred and sixty-nine patients had Ct reported for each of the genes on the first NPh swab, but only 271 also had an NPh swab performed 7 (±2) days later. Of these, 266 patients also had the Ct reported for each of the three genes on the T7 NPh swab. Thirteen patients were excluded because they were either treated with IL-6 inhibitors or the information was missing, reaching a final population of 253 patients. A flowchart of population selection is shown in Figure 1.

Characteristics of the enrolled patients are reported in Table 1. The median age in our cohort was 64 years (IQR 53–77), with a prevalence of males (64.8%). Overall, 85.4% of the enrolled population (216/253) had at least one comorbidity. Specifically, 57.3% (145/253) of the patients had cardiovascular diseases, 24.1% (61/253) had diabetes and 17.4% (44/253) were obese. Furthermore, 40.7% (103/253) of patients had severe COVID-19, requiring high-flux oxygen and non-invasive or invasive ventilation. The rate of patients admitted to the Intensive Care Unit (ICU) was 7.5% (19/253). The in-hospital mortality rate (non-survivors) was 14.3% (36/253). The majority of patients received concomitant corticosteroid treatment (150/253, 59.3%).

Sixty-seven patients (26.5%) received lopinavir or darunavir/ritonavir (PI group), 123 (48.6%) received remdesivir (R group) and 63 (24.9%) received no antiviral treatment (NT group) (Table 1). Remdesivir was started at a median of 9 days from symptoms’ onset (IQR 6–11), while PI was at a median of 6 days (IQR 3–10).

Patients were significantly younger in the NT group (median 55 vs. 64 vs. 67 years in NT, PI and R groups, respectively, *p* = 0.004). In the remdesivir group, males were predominant (*p* = 0.032) and more patients were obese (*p* = 0.006) and had cardiovascular diseases (*p* = 0.016) compared to the other groups. Patients in the remdesivir group had a higher percentage of severe COVID-19 (14.3% vs. 20.9% vs. 65% in NT, PI and R groups, respectively, *p* < 0.001) and were more frequently admitted to the ICU (0% vs. 7.5% vs. 11.4% in NT, PI and R group, respectively, *p* = 0.021). No differences in overall mortality were reported amongst the three groups (9.5% vs. 13.4% vs. 17.1% in NT, PI and R groups, respectively, *p* = 0.369). No significant differences were recorded in time from symptoms’ onset to T0 NPh swab collection amongst the three treatment groups (median days 6 (IQR 7.5–9.5) vs. 6 (IQR 2–10.5) vs. 7 (IQR 4–9.75) in NT, PI and R groups, respectively, *p* = 0.316). Significantly more patients in the remdesivir group received concomitant steroid treatment compared to the other groups (23.8% vs. 23.9% vs. 96.7% in NT, PI and R groups, respectively, *p* < 0.001) (Table 1).

When stratified by outcome, in survivors and non-survivors, median Ct values of the E, N and RdRP genes were significantly lower in the non-survivors group compared to survivors at both T0 and T7 (Table 2). Moreover, ROC analysis confirmed that Ct values were able to discriminate between survivors and non-survivors at both T0 and T7. After applying the Youden criterion, cut-off values for Ct E, N and RdRP were identified at T0 and T7 (T0: Ct E < 24.39 sensitivity 67% specificity 65%; Ct N < 24.33 sensitivity 69% specificity 64%; Ct RdRP < 23.10 sensitivity 56% specificity 74%. T7: Ct E < 26.96 sensitivity 61% specificity 83%; Ct N < 29.67 sensitivity 72% specificity 74%; Ct RdRP < 31.85 sensitivity 78% specificity 67%) (Figure 2).

Considering the previously identified cutoff by Kurzeder et al. [7] (Ct ≤ 26 on hospital admission), also in our cohort mortality rate was significantly increased in patients with Ct values below the cutoff at T0 (mortality rate for Ct ≤ 26 vs. Ct > 26: E 21.2% vs. 8.2%, *p* = 0.004; N 20.2% vs. 8.1% *p* = 0.007; RdRP 20.9% vs. 9.1% *p* = 0.01). Similarly, at T7, the mortality rate was significantly increased in patients with Ct ≤ 30 (mortality rate Ct ≤ 30 vs. Ct > 30: E 35.4% vs. 6.9%, *p* < 0.0001; N 38.2% vs. 5.4% *p* < 0.0001; RdRP 38.3% vs. 6.7% *p* < 0.0001).

As for viral decay, Ct variation was significantly reduced in the non-survivors group compared to survivors (delta CT E gene 5.15 vs. 7.29, *p* = 0.084; delta Ct N gene 4.48 vs. 6.60, *p* = 0.050; delta Ct RdRP gene 5.00 vs. 7.27, *p* = 0.018, in non-survivors vs. survivors, respectively, Table 2). As for steroid treatment, no differences were found in the median Ct values for all the three analyzed genes at T0 and T7 and in viral decay (T7-T0) between patients who received concomitant corticosteroids and who did not (Table 2).

No correlation was noted between age, comorbidities and lymphocyte subpopulation collected at T0 and NPh swab Ct for E, N and RdRP genes at T0 (Table 3). Ct variation directly correlated with CD3+ lymphocyte absolute counts collected at T0 (delta Ct E gene Spearman r = 0.138, *p* = 0.029; delta Ct N gene Spearman r = 0.124, *p* = 0.051; delta Ct RdRP gene Spearman r = 0.134, *p* = 0.034), and CD4+ (delta Ct E gene Spearman r = 0.140, *p* = 0.027; delta Ct N gene Spearman r = 0.147, *p* = 0.020; delta Ct RdRP gene Spearman r = 0.139, *p* = 0.028), with lower CT variation over time observed in patients who presented with more severe lymphopenia at T0 (Table 3). As for inflammatory markers assessed at ward admission (interleukin-6 (IL-6), C-reactive protein (CRP), D-dimers, fibrinogen), a direct correlation was found between D-dimer and fibrinogen and Ct at T0, while IL-6 was inversely correlated with T0 Ct of the three SARS-CoV-2 genes. No correlation was found for any of the inflammatory markers assessed at ward admission with delta Ct for E, N and RdRP genes of SARS-CoV-2 (Table 3).

Focusing on antiviral treatment, median Ct values of the N gene were significantly lower on both T0 and T7 NPh swabs in the remdesivir group, compared to the PI group (T0: 23.49 vs. 29.78 vs. 24.81 in NT, PI and R groups, respectively, *p* < 0.001; T7: 30.72 vs. 34.97 vs. 32.01 in NT, PI and R group, respectively, *p* = 0.019); median Ct values of the other genes did not differ amongst the three treatment groups, both at T0 and T7 (Table 4). As for viral decay, no significant differences were noted in the Ct variation of any of the analyzed genes throughout the three treatment groups (Table 4; Figure 3).

Given the greater percentage of severe patients in the remdesivir group, additional analyses were performed separately for non-severe and severe patients: no significant differences were noted in the Ct of the three genes at T0 and T7 nor in the Ct variation (T7-T0) of any of the analyzed genes, throughout the three treatment groups (Appendix A).

To further eliminate possible confounding factors, the analysis was restricted to a population of patients that received antiviral therapy (with either remdesivir or PI) within 10 days from symptoms’ onset (R started at a median of 7 (IQR 4–9) days, PI started at a median of 5 (IQR 2–7) days) and subjects who did not receive any antiviral, including a final population of 194 patients (Appendix A Appendix A).

Median Ct values of the three SARS-CoV-2 genes were not significantly different on either T0 and T7 NPh swabs in the different treatment groups (*p* > 0.05) (Appendix A Appendix A). As for viral decay, no significant differences were observed in the Ct variation of any of the analyzed genes throughout the three groups (Appendix A Appendix A). In addition, in this restricted population, considering the greater percentage of severe patients in the remdesivir group, additional statistical analyses were performed separately for non-severe and severe patients: no significant differences were noted in the Ct of the three genes at T0 and T7 nor in the Ct variation (T7–T0) of any of the analyzed genes, throughout the three treatment groups (Appendix A Appendix A).

## 4. Discussion

Our study shows no differences in SARS-CoV-2 Ct variation in NPh swabs collected over time between patients who received either remdesivir, protease inhibitors, or no direct antiviral treatment. Non-survivors showed lower Ct values at the RT-PCR for SARS-CoV-2 E, N and RdRP genes on the NPh swab collected at ID ward admission and a reduced Ct variation over time compared to survivors. Lower CD3+ and CD4+ T-lymphocyte absolute counts collected at T0 were correlated with reduced Ct variation over 7 days.

HIV protease inhibitors (lopinavir and darunavir, combined with ritonavir) have been used in the early SARS-CoV-2 pandemic phase. Randomized controlled trials showed no clinical benefit in time to clinical improvement or mortality reduction in patients treated with lopinavir/ritonavir compared to standard of care [16,18,29,30], and these drugs were briskly withdrawn from clinical practice [31]. Similarly, viral RNA loads evaluated over time did not differ between lopinavir/ritonavir and standard of care recipients [29].

In clinical trials, remdesivir has been effective in reducing time to recovery in patients hospitalized for COVID-19, but there is conflicting evidence about its efficacy in reducing hospital stay, disease severity and mortality [13,15,19,20,21,22,23,30,32,33,34]. In the latest COVID-19 treatment guidelines, remdesivir use has been recommended mainly in high-risk outpatients to reduce hospitalization [15,16,17,19]. In the inpatient setting, an effect on mortality and the need for mechanical ventilation has been reported, but still, there are conflicting data on the magnitude of remdesivir efficacy in treating severe COVID-19 [15,16,19,20,21,22,23]. Differences in the results might be due to different outcomes and severity definitions or population sizes. In a pandemic scenario, when global efforts are being made to collect evidence in a timely manner, sharing definitions for trial patients’ enrollment might favor high-quality literature production and help clinicians in patient management [35]. In our cohort, no differences in survival rates were recorded for patients treated with remdesivir, protease inhibitors, or no antiviral agents (*p* = 0.389), even if patients in the remdesivir group were more severe (*p* = <0.001) and more often admitted to the ICU (*p* = 0.021). It is not possible to exclude that the increased percentage of severe COVID-19 patients in the R group might have masked remdesivir clinical effectiveness in this population, compared to PI and NT groups. Nonetheless, the study was not designed to assess the clinical efficacy of remdesivir and its impact on mortality but to evaluate its effects on SARS-CoV-2 viral load decay in NPh swabs of COVID-19 patients.

One multicentric prospective randomized controlled trial conducted in China by Wang et al. evaluated viral decay in respiratory specimens (NPh swabs and sputum), showing no significant difference in viral load variation over time between patients receiving remdesivir or placebo [22]. Subsequent real-life studies show conflicting results on the remdesivir effect on viral decay [9,18,25,26,27,28,34,36,37]. A recent prospective controlled study involving 151 patients treated with either remdesivir plus dexamethasone or dexamethasone alone showed a significant reduction in overall mortality and a significantly increased viral clearance in the group treated with the association of remdesivir plus dexamethasone [34]. In a cohort of 45 COVID-19 patients (27 treated with remdesivir), Biancofiore et al. demonstrated an increased decay of Ct values in patients treated with remdesivir compared to standard of care (daily decrease: 0.61 vs. 0.33 cycles, respectively, *p* = 0.045) [26]. Similarly, in a cohort of 86 patients (48 treated with remdesivir), Joo et al. showed a steeper slope in Ct values variation over time (5.1 vs. 2.68 cycles, respectively, *p* = 0.007) in COVID-19 severe patients who received remdesivir compared to patients that did not receive remdesivir [27]. Conversely, Goldberg et al., in a cohort of 142 COVID-19 hospitalized patients (29 treated with remdesivir), failed to demonstrate an effect of remdesivir in reducing SARS-CoV-2 N gene Ct values on NPh swabs more rapidly than in untreated patients. Accordingly, our study showed no effect of remdesivir on Ct variation over time when compared to protease inhibitors or no treatment (*p* > 0.2 for all the three analyzed RT-PCR SARS-CoV-2 target genes). Compared to previous studies, we analyzed a larger cohort of COVID-19 patients with a relevant proportion of remdesivir-treated patients.

Trying to eliminate confounding factors, our study population was restricted to COVID-19 patients who received antivirals (either remdesivir or PIs) within 10 days from symptoms’ onset and those who did not receive any antiviral, but again no differences in Ct variation were recorded amongst the treatment groups (*p* > 0.2 for all the three analyzed RT-PCR SARS-CoV-2 target genes). We furthermore tried to reduce factors that could alter the immune response to SARS-CoV-2, excluding patients that received IL-6 inhibitors, but corticosteroid use was widely variable in the cohort, both regarding drug type and dosages, due to the important changes in official recommendations that occurred during the study period [31]. In our cohort, as expected, the majority of patients treated with remdesivir received concomitant corticosteroid treatment in a much higher percentage than in the PI and NT groups (*p* < 0.001). Nonetheless, corticosteroid use seems not to have influenced Ct variation over time in the overall population. Discrepancies between remdesivir clinical and virological effects trigger questions about the pathophysiology of COVID-19, the role of host response in clinical evolution and the possible effects of remdesivir on SARS-CoV-2 in compartments other than the respiratory system.

To date, wide gaps remain in understanding COVID-19 pathophysiology, including viral and host role in the disease severity. In the early pandemic, some demographic factors and pre-existing conditions have been shown to increase the risk of severe COVID-19 [3,38,39]. Later, it was hypothesized that the host immunity response contributes greatly to the infection’s severity [40,41,42,43,44]. The alteration of some inflammatory markers, such as D-dimer and C reactive protein and lymphopenia early after SARS-CoV-2 infection, might help identify patients at higher risk of severe COVID-19 and unfavorable outcomes [45,46]. Viral load influence on clinical manifestation is less clear, with conflicting results [5,46,47,48,49,50,51].

Interestingly, a significant correlation has been consistently reported between viral load, either evaluated directly or through Ct, and laboratory inflammatory markers and lymphopenia collected at hospital admission [5,46,47,48,52]. In our study, low Ct on the admission NPh swab, accounting for a greater viral load, was associated with an increased rate of in-hospital deaths (*p* < 0.002 for all the three analyzed RT-PCR SARS-CoV-2 targets). No correlation was found between T0 NPh swab Ct with other factors that might have had an impact on COVID-19 severity and outcome, such as age and comorbidity. As for inflammatory markers, a low CD3+ and CD4+ lymphocyte count at admission significantly correlated with lower viral decay on the NPh swab. An indirect correlation with IL-6 and T0 Ct of the E and RdRP genes were found, probably accounting for a greater cytokine storm triggered by a higher viral load, but no correlation was found with subsequent viral load decay. It could be speculated that an initial higher viral load might trigger an abnormal immune host response, with later clinical manifestation driven by immune dysfunction rather than by direct viral action. In this scenario, the precocious use of antiviral treatment might reduce immunological host response, mitigating COVID-19 severity. Immunocompromised hosts represent a unique population to investigate virus–host interactions: there are case reports describing SARS-CoV-2 viral load reduction in sputum after treatment with remdesivir even with persistently negative NPh swabs [53] and plasma viral load reduction in a patient with persistently positive NPh swabs and viable virus isolation on VERO cellular culture [54].

As for viral clearance, even less is known about SARS-CoV-2 load variations in the respiratory tract and their correlation with clinical disease course, but severe COVID-19 seems to be associated with a slower viral clearance [52,55,56]. In our cohort, a smaller Ct variation was recorded for the N and the RdRP genes in non-survivors compared to survivors (*p* ≤ 0.05), irrespective of the treatment received. Furthermore, greater lymphopenia on admission was significantly correlated with lower Ct variation over time, showing a correlation between immune status at infection onset and the ability to clear the virus from respiratory specimens. A recently published article by our research group showed that lymphopenia at ID ward admission was correlated with COVID-19 severity and poor outcome [42]; whether or not reduced viral clearance from the upper respiratory tract is associated with worse COVID-19 outcomes is yet to be established. An interesting study on rhesus macaques showed that remdesivir significantly reduced SARS-CoV-2 viral load in the lungs preventing pneumonia development, even if no changes in viral quantity were observed in nasal and throat specimens [57]. It is easy, in clinical practice, to extrapolate data on NPh swab viral Ct, given their role in SARS-CoV-2 infection diagnosis: being able to correlate NPh swabs Ct values with viral load in other specimens (e.g., lower respiratory tract specimens) or other body compartments and correlate them with more complex clinical outcomes, might allow delving into viral–host interaction during SARS-CoV-2 infection and optimize COVID-19 clinical management. There are relatively few studies focusing on extra-respiratory compartments, such as blood, even if COVID-19 is being increasingly recognized as a systemic disease, with endothelial dysfunction and multi-organ failure in the most severe cases. Recent evidence indicates a higher plasmatic viremia in more severe patients and its association with adverse outcomes and with inflammatory markers alterations [24,47,54,58]. A recently published observational study analyzes the effect of remdesivir treatment on viral load decay in respiratory specimens (NPh swabs, oropharyngeal swabs, sputum) and blood samples [25]. The study reports a significant viral decay in hospitalized patients treated with remdesivir (18 patients) compared with patients who received no antiviral treatment (33 patients), with the majority of data used in the analysis of viral load reduction derived from blood samples (45.7%). Wider high-quality studies are needed to assess antiviral drug effects on SARS-CoV-2 viremia and disease outcome.

This study has several limitations; firstly, the methodology flaws associated with the retrospective study design. It is a single-center study hence findings cannot easily be generalized. Viral decay was assessed indirectly by evaluating the cycle threshold on RT-PCR for SARS-CoV-2. Furthermore, viral decay was evaluated only in upper respiratory tract specimens, which could be dissimilar from viral load in the lungs. SARS-CoV-2 isolated strains were not genotyped, and variants’ impact on different outcomes could not be assessed; however, in Italy during the enrollment period, the presence of variants different from the ancestral was considered to be negligible, as its impact on the used diagnostic tests, which do not detect the SARS-CoV-2 protein [59,60]. Finally, the time from symptoms’ onset to the first NPh swab was variable throughout the enrolled population, even if the differences were not statistically significant in the different treatment groups; additionally, the use of delta Ct (T7-T0) should have normalized differences in initial Ct values.

## 5. Conclusions

Viral decay of SARS-CoV-2 RNA on NPh swabs of hospitalized COVID-19 patients did not differ in subjects treated with remdesivir and protease inhibitors and was similar to that observed in patients who did not receive any antiviral therapy. Further studies are needed to assess the specific antiviral effect of remdesivir on COVID-19 patients and its impact on disease severity and outcome.

## Figures and Tables

**Figure 1 microorganisms-11-00312-f001:**
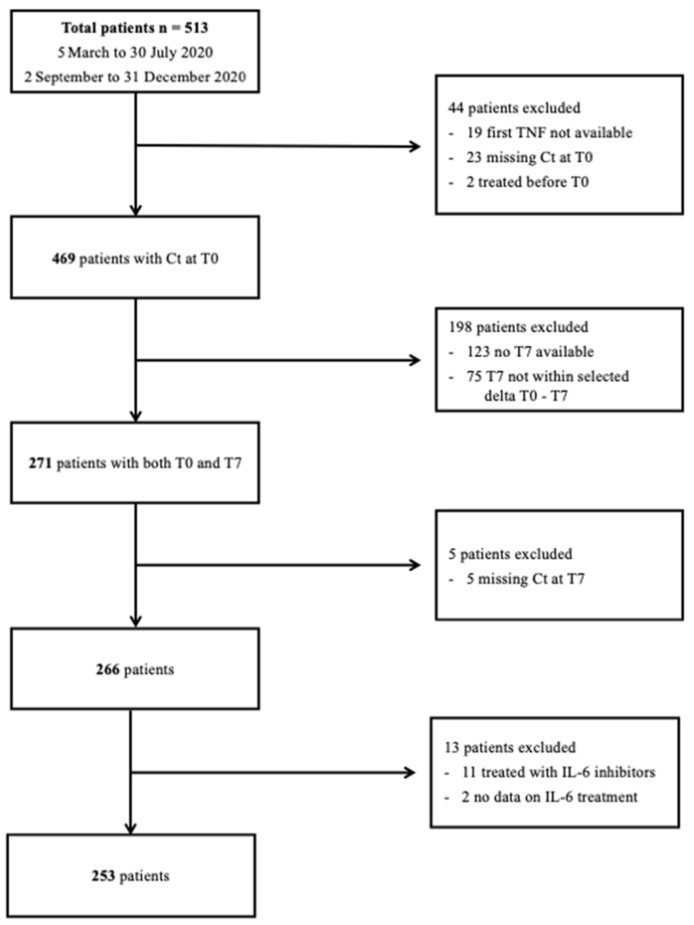
Population selection criteria.

**Figure 2 microorganisms-11-00312-f002:**
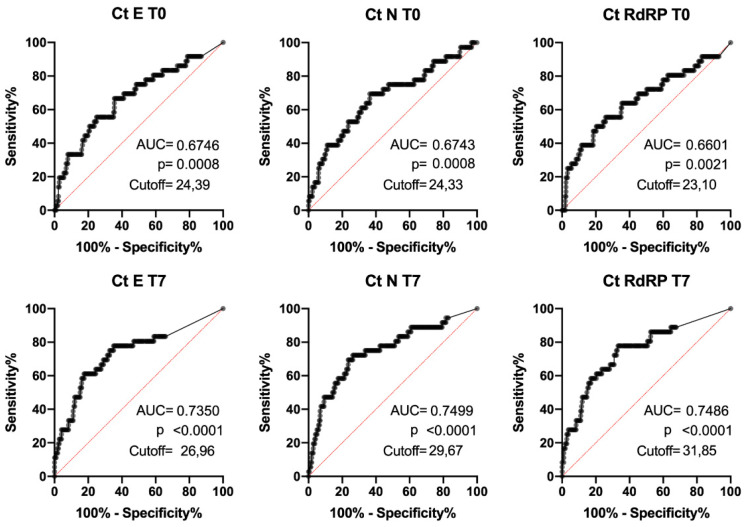
ROC curves for SARS-CoV-2 Real-Time PCR Ct values of E, N and RdRP genes on nasopharyngeal swabs at T0 and T7. Area under the curve (AUC), *p*-values and cutoff values obtained after applying Youden’s criterion are reported in each panel.

**Figure 3 microorganisms-11-00312-f003:**
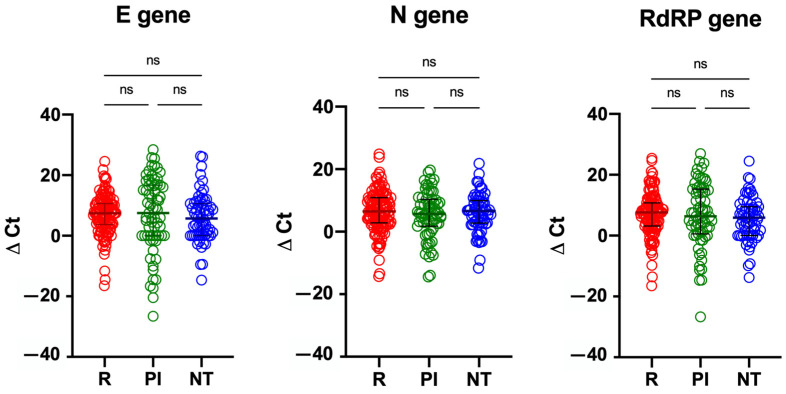
Ct variation for SARS-CoV-2 E, N and RdRP genes in patients divided by the antiviral treatment received. Horizontal lines represent medians; whiskers represent interquartile ranges. NT: no antiviral treatment; PI: protease inhibitors; R: remdesivir.

**Table 1 microorganisms-11-00312-t001:** General characteristics of the study population, overall and divided by treatment groups (remdesivir, protease inhibitors and no antiviral treatment).

	Overall Population (N = 253)	Remdesivir (R) (N = 123; 48.6%)	Protease Inhibitors (PI) (N = 67; 26.5%)	No Treatment (NT) (N = 63; 24.9%)	*p*
Age: median (IQR)	64 (53–77)	67 (58.5–76.5)	64 (51–81.5)	55 (44.5–75.5)	0.004
Sex: M/F	164/89 (64.8/35.2)	89/34 (72.4/27.6)	36/31 (53.7/46.3)	39/24 (61.9/38.9)	0.032
Time from symptoms’ onset to T0 NPhS: median (IQR) *	7 (3–10)	7 (4–9.75)	6 (2–10.50)	6 (7.50–9.50)	0.316
Non-severe/severe	150/103 (59.3/40.7)	43/80 (35/65)	53/14 (79.1/20.9)	54/9 (85.7/14.3)	<0.001
ICU admission	19 (7.5)	14 (11.4)	5 (7.5)	0	0.021
Survivors/non-survivors	217/36 (85.7/14.3)	102/21 (82.9/17.1)	58/9 (86.6/13.4)	57/6 (90.5/9.5)	0.369
Corticosteroid treatment	150 (59.3)	119 (96.7)	16 (23.9)	15 (23.8)	<0.001
Comorbidities					
Any	216 (85.4)	108 (87.8)	55 (82.1)	53 (84.1)	0.538
Obesity	44 (17.4)	31 (25.2)	7 (10.4)	6 (9.5)	0.006
Cardiovascular	145 (57.3)	81 (65.8)	36 (53.7)	28 (44.4)	0.016
Diabetes	61 (24.1)	36 (29.3)	13 (19.4)	12 (19.1)	0.175
Endocrinologic	32 (12.6)	16 (13)	9 (13.4)	7 (11.1)	0.911
Cerebrovascular	21 (8.3)	14 (11.4)	4 (5.9)	3 (4.7)	0.218
Chronic viral hepatitis	3 (1.2)	1 (0.8)	1 (1.5)	1(1.5)	0.867
Pulmonary	30 (11.8)	18 (14.6)	8 (11.9)	4 (6.4)	0.255
Renal	19 (7.5)	3 (2.4)	6 (8.9)	10 (15.8)	0.004
Solid Tumor	34 (13.4)	18 (14.6)	11 (16.4)	5 (7.9)	0.316
Hematologic	21 (8.3)	11 (8.9)	7 (10.4)	3 (4.7)	0.470
Neurologic/Psychiatric	43 (16.9)	13 (10.6)	15 (22.3)	15 (23.8)	0.029
Rheumatologic	17 (6.7)	8 (6.5)	3 (4.5)	6 (9.5)	0.513
Other	49 (19.3)	19 (15.4)	15 (22.3)	15 (23.8)	0.302

Quantitative data are presented as median (IQR); qualitative data are presented as absolute frequency (percentage). * Data for 236 patients, 17 asymptomatic patients were excluded from this analysis. ICU: Intensive Care Unit; IQR: interquartile range; NPhS: nasopharyngeal swab.

**Table 2 microorganisms-11-00312-t002:** SARS-CoV-2 viral parameters of the study population, divided by outcome (survivors vs. non-survivors) and concomitant corticosteroid treatment (corticosteroid vs. no corticosteroid).

	Survivors (N = 217; 85.7%)	Non-Survivors (N = 36; 14.3%)	*p*	Corticosteroid (N = 150; 59.3%)	No Corticosteroid (N = 103; 40.7%)	*p*
T0 NPh swab						
Median Ct E	26.87 (22.14–32.03)	21.48 (17.27–26.75)	<0.001	26.74 (22.17–31.11)	25.19 (19.07–33.15)	0.364
Median Ct N	26.52 (21.94–31.62)	21.55 (16.76–26.92)	<0.001	25.56 (21.48–30.45)	26.71 (20.45–33.69)	0.361
Median Ct RdRP	27.41 (22.68–32.46)	22.25 (17.56–28.95)	0.002	27.26 (22.62–32.05)	26.39 (20.54–32.65)	0.390
T7 NPh swab						
Median Ct E	34.44 (29.26–45.00)	25.20 (20.88–31.51)	<0.001	33.51 (28.44–37.90)	34.14 (26.66–45.00)	0.277
Median Ct N	33.34 (28.88–37.39)	25.25 (21.21–31.13)	<0.001	32.07 (26.62–36.67)	33.50 (27.71–38.17)	0.129
Median Ct RdRP	34.98 (30.09–45.00)	26.07 (21.60–31.56)	<0.001	33.82 (28.91–45.00)	33.80 (27.69–45.00)	0.842
Viral decay (T7-T0)						
Median ΔCt E	7.29 (1.50–12.03)	5.15 (−0.35–9.12)	0.084	7.10 (2.44–10.47)	7.79 (0.00–14.78)	0.497
Median ΔCt N	6.60 (2.83–10.90)	4.48 (−0.96–8.58)	0.050	6.43 (2.80–10.18)	5.98 (2.64–10.91)	0.937
Median ΔCt RdRP	7.27 (2.67–11.98)	5.00 (−1.53–8.43)	0.018	6.95 (2.80–10.32)	7.08 (0.93–13.51)	0.817

Data are reported as median [IQR]. The analysis was performed using both parametric and non-parametric statistical tests, which provided similar results (more details about the statistical tests and the effect size can be found in Appendix A); *p*-values obtained with the Mann–Whitney test for nonparametric independent variables are shown in the table. Ct: cycle threshold; E: Envelope gene; IQR: interquartile range; N: Nucleocapsid gene; NPh: nasopharyngeal swab; RdRP: RNA-dependent RNA Polymerase gene.

**Table 3 microorganisms-11-00312-t003:** Correlation between age, comorbidity score and laboratory parameters with the Ct of the three SARS-CoV-2 genes on NPh swab at T0 and with Ct variation.

	Median Ct E T0	Median CT N T0	Median Ct RdRP T0	Median Delta Ct E	Median Delta CT N	Median Delta Ct RdRP
Age	−0.060 *p* = 0.339	−0.092 *p* = 0.143	−0.063 *p* = 0.320	−0.060 *p* = 0.344	−0.075 *p* = 0.235	−0.084 *p* = 0.185
Comorbidity score	−0.095 *p* = 0.133	−0.122 *p* = 0.053	−0.090 *p* = 0.152	−0.022 *p* = 0.731	−0.042 *p* = 0.508	−0.038 *p* = 0.542
IL−6	−0.132 *p* = 0.047	−0.106 *p* = 0.112	−0.131 *p* = 0.048	0.023 *p* = 0.726	0.010 *p* = 0.883	−0.006 *p* = 0.923
D-dimer	0.170 *p* = 0.011	0.102 *p* = 0.127	0.161 *p* = 0.016	−0.127 *p* = 0.057	−0.025 *p* = 0.713	−0.110 *p* = 0.100
CRP	0.063 *p* = 0.335	0.034 *p* = 0.602	0.081 *p* = 0.213	−0.004 *p* = 0.952	−0.0008 *p* = 0.990	−0.014 *p* = 0.826
Fibrinogen	0.160 *p* = 0.016	0.150 *p* = 0.024	0.147 *p* = 0.027	0.036 *p* = 0.588	0.039 *p* = 0.555	0.055 *p* = 0.406
Lymphocyte total count (T0)	0.032 *p* = 0.620	0.084 *p* = 0.193	0.037 *p* = 0.570	0.128 *p* = 0.047	0.092 *p* = 0.153	0.108 *p* = 0.092
Lymphocyte subpopulation (T0)						
CD3+ #	0.042 *p* = 0.512	0.099 *p* = 0.120	0.051 *p* = 0.425	0.138 *p* = 0.029	0.124 *p* = 0.051	0.134 *p* = 0.034
CD3 + CD4+ #	0.046 *p* = 0.472	0.102 *p* = 0.109	0.056 *p* = 0.379	0.140 *p* = 0.027	0.147 *p* = 0.020	0.139 *p* = 0.028
CD3 + CD8+ #	0.034 *p* = 0.595	0.087 *p* = 0.171	0.036 *p* = 0.571	0.120 *p* = 0.058	0.094 *p* = 0.138	0.124 *p* = 0.050
CD19+ #	0.188 *p* = 0.003	0.198 *p* = 0.002	0.179 *p* = 0.005	0.001 *p* = 0.958	0.023 *p* = 0.713	0.016 *p* = 0.795
CD3-CD16 + CD56+ #	−0.099 *p* = 0.118	−0.053 *p* = 0.408	−0.078 *p* = 0.223	0.049 *p* = 0.440	0.031 *p* = 0.628	0.028 *p* = 0.664
CD4/CD8 ratio	0.039 *p* = 0.544	0.023 *p* = 0.713	0.037 *p* = 0.563	−0.005 *p* = 0.940	0.059 *p* = 0.352	0.007 *p* = 0.918

Data are represented as Spearman’s rho coefficient and *p*-value. Statistically significant correlations are highlighted in bold. Comorbidity score: sum of the number of comorbidities for each patient; CRP: C reactive protein; Ct: cycle threshold; E: Envelope gene; IL-6: interleukin-6; N: Nucleocapsid gene; NPh swab: nasopharyngeal swab; RdRP: RNA-dependent RNA Polymerase gene; #: absolute count.

**Table 4 microorganisms-11-00312-t004:** SARS-CoV-2 viral parameters of the study population, divided by treatment group (remdesivir, protease inhibitors and no antiviral treatment).

	All Patients (N= 253)	Remdesivir (R) (N = 123; 48.6%)	Protease Inhibitors (PI) (N = 67; 26.5%)	No Treatment (NT) (N = 63; 24.9%)	*p*
T0 NPh swab					
Median Ct E	26.34 (21.09–31.53)	26.26 (22.12–30.3)	27.52 (22.42–34.94)	24.4 (18.62–34.54)	0.159
Median Ct N	25.97 (21.30–30.83)	24.81 (21.43–29.6)	29.78 (24.95–34.49)	23.49 (18.69–33.45)	<0.001
Median Ct RdRP	21.17 (21.97–32.20)	27.07 (22.59–31.62)	28.02 (23.88–32.72)	25.88 (19.00–34.76)	0.392
T7 NPh swab					
Median Ct E	33.73 (27.17–45.00)	33.79 (29.12–37.7)	45.00 (28.00–45.00)	31.47 (23.895–45.00)	0.078
Median Ct N	32.92 (27.01–37.09)	32.01 (27.14–35.11)	34.97 (30.47–39.04)	30.72 (25.15–37.59)	0.019
Median Ct RdRP	33.80 (28.49–45.00)	34.31 (29.75–38.98)	34.95 (29.46–45.00)	32.88 (25.00–45.00)	0.280
Viral decay (T7-T0)					
Median Ct E	7.25 (1.25–11.78)	7.46 (3.71–10.61)	7.53 (0.00–16.65)	5.72 (0.00–10.52)	0.357
Median Ct N	6.42 (2.65–10.43)	6.51 (2.85–10.82)	5.73 (2.20–10.23)	6.60 (2.78–9.89)	0.646
Median Ct RdRP	7.02 (2.16–11.26)	7.66 (3.33–10.77)	6.37 (0.93–15.14)	5.91 (0.00–9.37)	0.233

Data reported as median (IQR). Ct: cycle threshold; E: Envelope gene; IQR: interquartile range; N: Nucleocapsid gene; NPh: nasopharyngeal swab; RdRP: RNA-dependent RNA Polymerase gene.

## Data Availability

Data are deposited in an ad hoc created Excel database, available on request from the corresponding author. The data are not publicly available due to privacy and ethical restrictions.

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
