# Peer review of "Remdesivir Influence on SARS-CoV-2 RNA Viral Load Kinetics in Nasopharyngeal Swab Specimens of COVID-19 Hospitalized Patients: A Real-Life Experience"

_microorganisms, 2023, doi:10.3390/microorganisms11020312_

Round 1

Reviewer 1 Report (Previous Reviewer 1)

The manuscript describes a study comparing viral load in patients treated with remdesivir to viral load in patients receiving other treatment. I just have a few questions that the authors can comment on.

- An analysis of remdesivir treatment in macaques (Dobrovolny, Virology 2020) found that the only statistically significant difference between treated and untreated animals was a change in the viral decay rate. Can the authors comment on possible reasons for the different finding here.

- While the authors do mention that differences in timing of when the viral load samples were taken (relative to start of infection) might affect their results, can they also comment on how the timing of the start of remdesivir treatment might be affecting their results? Remdesivir needs to be provided as early as possible during the infection and late treatment is known to have little effect. How might that have played a role here?

Author Response

Reviewer 1

Comments and Suggestions for Authors

The manuscript describes a study comparing viral load in patients treated with remdesivir to viral load in patients receiving other treatment. I just have a few questions that the authors can comment on.

- An analysis of remdesivir treatment in macaques (Dobrovolny, Virology 2020) found that the only statistically significant difference between treated and untreated animals was a change in the viral decay rate. Can the authors comment on possible reasons for the different finding here.

Answer:

We thank the reviewer for this observation. Dobrovolny et al examined experimental data from Williamson et al.  (Nature 2020) who performed a study on the effectiveness of remdesivir treatment of SARS-CoV-2 in rhesus macaques. In their study, Williamson et al found that despite a reduction in virus replication in the lower respiratory tract, neither viral loads nor infectious virus titers were reduced in nose, throat or rectal swabs collected from remdesivir-treated animals, except a significant difference in virus titer in throat swabs collected on day 1 post-infection and in viral loads in throat swabs collected on day 4 post-infection.

Based on these data, Dobrovolny et al developed a mathematical model to describe SARS-CoV-2 viral kinetics. However, considering some limitations of the study, such as the small number of animals used in Williamson’s experiments, the fact that viral decay was arbitrarily fixed, the authors recommend being cautious in generalizing their mathematical model of SARS-CoV-2 viral decay. Furthermore, Dobrovolny’s mathematical model suggests that the application of remdesivir can lengthen SARS-CoV-2 infections.

Conversely, our results are in line with the original data obtained by Williamson et al. in the upper respiratory tract, showing no differences in viral decay after comparing remdesivir treated vs untreated rhesus macaques. We discussed this point in lines 388-391:    

“An interesting study on rhesus macaques showed that remdesivir significantly reduced SARS-CoV-2 viral load in lungs preventing pneumonia development, even if no changes in viral quantity were observed in nasal and throat specimens [59]”

- While the authors do mention that differences in timing of when the viral load samples were taken (relative to start of infection) might affect their results, can they also comment on how the timing of the start of remdesivir treatment might be affecting their results? Remdesivir needs to be provided as early as possible during the infection and late treatment is known to have little effect. How might that have played a role here?

Answer:

We thank the reviewer for this comment, and we believe that the timing of remdesivir administration is of paramount importance to understand its effect on viral kinetics. To this purpose, we added a sentence in the results section describing the timing of remdesivir and protease inhibitors start from symptoms onset (line 161-162):

“Remdesivir was started at a median of 9 days from symptoms’ onset (IQR 6-11), while PI at a median of 6 days (IQR 3-10).”

Furthermore, we performed a subanalysis (supplementary material S3, S4 and S5) including only those patients who started either remdesivir or protease inhibitors within 10 days from symptoms onset (according to the Italian Medicines Agency recommendations for remdesivir use in COVID-19 patients). This analysis did not show any statistically significant difference of SARS-CoV-2 viral decay in nasopharyngeal swabs, after comparing the three treatment groups (R vs PI vs NT) (lines 266-279):

“To further eliminate possible confounding factors, the analysis was restricted to a population of patients that received antiviral therapy (with either remdesivir or PI) within 10 days from symptoms’ onset (R started at a median of 7 [IQR 4-9] days, PI started at a median of 5 [IQR 2-7] days) and subjects who did not receive any antiviral, including a final population of 194 patients (supplementary material table S3).

Median Ct values of the three SARS-CoV-2 genes were not significantly different on either T0 and T7 NPh swabs in the different treatment groups (p>0.05) (supplementary material table S4). As for viral decay, no significant differences were observed in the Ct variation of any of the analyzed genes throughout the three groups (supplementary material table S4). Also in this restricted population, considering the greater percentage of severe patients in the remdesivir group, additional statistical analyses were performed separately for non-severe and severe patients: no significant differences were noted in the Ct of the three genes at T0 and T7 nor in the Ct variation (T7-T0) of any of the analyzed genes, throughout the three treatment groups (supplementary material table S5).”

Reviewer 2 Report (Previous Reviewer 3)

The authors have provided satisfactory responses to the comments.

Author Response

Reviewer 2

Comments and Suggestions for Authors

The authors have provided satisfactory responses to the comments.

Answer:

We thank the reviewer for the comment.

This manuscript is a resubmission of an earlier submission. The following is a list of the peer review reports and author responses from that submission.

Round 1

Reviewer 1 Report

The manuscript describes a retrospective study of COVID-19 patients treated with remdesivir, another antiviral, or not treated at all. The purported aim of the study is to determine the effect of remdesivir on the outcomes of COVID-19, but I am concerned that the analysis is not done properly, so it's not clear what the study adds to our knowledge of the effects of remdesivir.

- The introduction provides very minimal background of the studies performed using remdesivir. It also does not provide an overview of other COVID-19 treatments and where remdesivir currently stands in hierarchy of treatment options. 

- Given the clustering of the data and the large overlap of IQR in Figure 2, I find it hard to believe that there is a statistically significant difference between Ct in survivors and non-survivors --- the distributions are almost identical. I compare this to figure 3 which also has distributions that look very much the same, although the finding there is that there is no statistically significant difference. I don't know if the authors are not using the right statistical test, if it is implemented incorrectly, or if it is simply overpowered, but the results don't seem correct.

- The authors find no difference in mortality between the three groups, but I don't think they've accounted for the fact that patients who received remdesivir had more severe illness and thus were more likely to die if not treated.

- The abstract says the study data was collected through December 2021, but in the methods they list the collection as running through December 2020.

Author Response

Question 

- The introduction provides very minimal background of the studies performed using remdesivir. It also does not provide an overview of other COVID-19 treatments and where remdesivir currently stands in hierarchy of treatment options.

Answer

We thank the reviewer for the comment.

More detailed data on in vivo studies on remdesivir are reported in the discussion, allowing the introduction to be more focused on presenting the study rationale.

To clarify remdesivir position in the hierarchy of COVID-19 treatment options, the following paragraph has been added in the introduction, with the appropriate references updated in the manuscript.

 “Up to date remdesivir is recommended as early treatment (short 3 days treatment course) in fragile subjects, at high-risk for hospitalization, together with the oral antivirals nilmatrevir/ritonavir and molnupiravir. Remdesivir is also the only intravenous direct antiviral recommended to treat both non-severe and severe COVID-19 hospitalized patients (5 to 10 days treatment)”.

Question 

- Given the clustering of the data and the large overlap of IQR in Figure 2, I find it hard to believe that there is a statistically significant difference between Ct in survivors and non-survivors the distributions are almost identical. I compare this to figure 3 which also has distributions that look very much the same, although the finding there is that there is no statistically significant difference. I don't know if the authors are not using the right statistical test, if it is implemented incorrectly, or if it is simply overpowered, but the results don't seem correct.

Answer

We thank the reviewer for this comment. As reported in the methods section, statistical analysis for independent non-parametric variables were conducted using the non-parametric Mann-Whitney U test (two groups, continuous variable). Given the partial overlap of IQRs, to make the reading clearer, the numerical data used to design figure 2 and figure 3 are showed in table 2 and table 4 respectively, placed near to the figures.

To confirm the presented data and statistical differences, statistical analysis was repeated with the aforementioned test (Mann-Whitney U test), and confirmed the results reported in the manuscript. An additional statistical analysis was performed, using the Student t test for parametric continuous variables, confirming the presented results.

Question 

- The authors find no difference in mortality between the three groups, but I don't think they've accounted for the fact that patients who received remdesivir had more severe illness and thus were more likely to die if not treated.

Answer

We thank the reviewer for pointing this out, nonetheless assessing antiviral treatment effect on mortality was not included in the objectives of the study, hence statistical analysis was not intended to assess this outcome. The aim of the study was to investigate differences in viral decay in nasopharyngeal swabs in COVID-19 patients, after stratification according to the antiviral treatment received. We report the absence of a significant difference in gross numbers of deaths in the three treatment groups, but confounders factors were not assessed, so it cannot be inferred whether the different treatment received could have influenced the mortality.

To avoid confusion, the sentence “Aim of this study was to assess the effect of remdesivir on viral decay on NPh swab, evaluated through Ct variation over time, and on COVID-19 severity and mortality in hospitalized patients, treated with this antiviral, compared with patients who received protease inhibitors or no antiviral treatment” has been rewritten as follows

Aim of this study was to assess the effect of remdesivir on viral decay on NPh swab, evaluated through Ct variation over time in hospitalized patients, treated with this antiviral, compared with patients who received protease inhibitors or no antiviral treatment”.

Question 

- The abstract says the study data was collected through December 2021, but in the methods they list the collection as running through December 2020.

 Answer

We thank the reviewer for noticing the mistake. The dates have been corrected in the abstract (March 2020 – December 2020).

Reviewer 2 Report

The manuscript by Campogiani et al. describes a retrospective study in which a cohort of hospitalised patients has been stratified to determine the impact of antiviral treatment on SARS-CoV-2 viral load as estimated by the quantitation of viral RNA on nasal swabs collected from patients at admission time. 

The study concludes that no differences are observed between the different groups, regarding the antiviral treatment. Incidentally, and after additional stratification of the patients, some weak correlations are observed in terms of surrogate viral load (Ct) and some of the cellular and biochemical parameters analysed. 

The conclusions are exclusively based on statistical significance, which is good, but describing phenomena that may not be biologically or clinically meaningful given the limitations of the study, as acknowledged by the authors.

Author Response

We thank the reviewer for the appreciation of the manuscript and the comments.

Reviewer 3 Report

The authors analyzed the effect of remdesivir on viral decay, which was assessed through PCR Ct values on NPh swab of 253 COVID-19 patients. Methodology and data analysis looks correct and the results are useful as a future reference.

Comments for the authors:

Major points:

1.    Please include the prevalence of variants between March and December 2021 in Rome.

2.    Line 80: Please include the sensitivity of each assay against variants.

Minor points:

1.    Figures 2 & 3: Please delete 2A & 2B in the graph.

Author Response

Major points:

Question 

  1. Please include the prevalence of variants between March and December 2021 in Rome.

 Answer

We thank the reviewer for this comment.

The first SARS-CoV-2 variant (B.1.1.7, alpha variant) has been identified in September 2020 in the UK. In Italy, some studies report the prevalence of the alpha variant as around 3.5% at the end of December 2020. Given that our study enrolled patients from March 2020 to December 2020, we can assume that the presence of variants different from the ancestral one was rather low [Lai A. et al. Circulating SARS-CoV-2 variants in Italy, October 2020–March 2021. Virol J. 2021; 18: 168.].  National genotyping of SARS-CoV-2 isolated strains was implemented since the beginning of 2021, so strains analysis in our enrolled population was not available [https://www.epicentro.iss.it/coronavirus/sars-cov-2-monitoraggio-varianti-rapporti-periodici. Website, accessed 24th October 2022].

To include these considerations in the manuscripts, the following sentence has been added in the “limitations” section of the paper:

 “SARS-CoV-2 isolated strains were not genotyped, and variants impact on different outcomes could not be assessed; however, in Italy during the enrollment period the presence of variants different from the ancestral was considered to be negligible, as its impact on the used diagnostic tests, which do not detect the S SARS-CoV-2 protein”.

Question 

  1. Line 80: Please include the sensitivity of each assay against variants.

 Answer

We thank the reviewer for the observation.

As reported in the method section of the manuscript, all enrolled patients were tested for SARS-CoV-2 using the GeneFinderTM COVID-19 Plus RealAmp Kit, ELITech AllplexTM 2019-nCoV Assay (Seegene). The test targets the E (envelope), N (nucleocapsid) and RdRP (RNA-dependent RNA-Polymerase) genes, whilst it does not detect the Spike (S) protein of SARS-CoV-2, the region that mutates in the different SARS-CoV-2 variants, hence its diagnostic efficacy should not be hampered by viral variants.   

There is not much available literature on diagnostic sensitivity towards different SARS-CoV-2 variants; the published data we found did not describe any diagnostic variability for different variants with the test used at our hospital [Bozidis P. et al Unusual N Gene Dropout and Ct Value Shift in Commercial Multiplex PCR Assays Caused by Mutated SARS-CoV-2 Strain. Diagnostics (Basel). 2022 Apr 13;12(4):973]. Finally, as discussed in the previous question, we consider it can be assumed that in our enrolled population, the vast majority of patients were infected with the ancestral SARS-CoV-2 viral strain.

To include these considerations in the manuscripts, the following sentence has been added in the “limitations” section of the paper:

“SARS-CoV-2 isolated strains were not genotyped, and variants impact on different outcomes could not be assessed; however, in Italy during the enrollment period the presence of variants different from the ancestral was considered to be negligible, as its impact on the used diagnostic tests, which do not detect the S SARS-CoV-2 protein”.

Minor points:

Question 

  1. Figures 2 & 3: Please delete 2A & 2B in the graph.

 Answer

We thank the reviewer for noticing the typographical error. The figures have been changed in the manuscript.

Round 2

Reviewer 1 Report

My first comment on the deficiencies in the introduction was addressed by adding a paragraph to the discussion. So this comment was not addressed appropriately. I will also point out that although it appears as though both the introduction and discussion were completely re-written, they were not --- the "new" text is identical to the original text. My biggest issue with the manuscript was also not really addressed. The statistical analysis just doesn't match up with the graphs of the data. The authors claimed they re-did the tests --- if it was done improperly before, they could still have done it improperly again and gotten the same result. They claim that they also used a t-test, but don't present those results. Did they check if they were over-powered? They could also make the data available to have someone else check the statistical analysis. As it stands, they have not provided a reasonable explanation for why their analysis and graphs don't agree.